

# Development of a 5-mRNAsi-related gene signature to predict the prognosis of colon adenocarcinoma

Haifu Huang[1,*], Lin Lu[1,*], Yaoxuan Li[1], Xiumei Chen[1], Meng Li[1], Meiling Yang[1] and Xuewu Huang[2]

[1] Department of Hematology and Oncology, Shenzhen Hospital of Guangzhou University of Traditional Chinese Medicine, Shenzhen, China
[2] Tumor Center, The First Affiliated Hospital of Guangzhou University of Traditional Chinese Medicine, Guangzhou, China
* These authors contributed equally to this work.

Corresponding author
Xuewu Huang, drhuangxw@163.com

## ABSTRACT

**Aim.** To create a prognosis model based on mRNA-based stem index (mRNAsi) for evaluating the prognostic outcomes of colon adenocarcinoma (COAD).

**Background.** Generation of heterogeneous COAD cells could be promoted by the self-renewal and differentiation potential of cancer stem cells (CSCs). Biomarkers contributing to the development of COAD stem cells remained to be discovered.

**Objective.** To develop and validate an mRNAsi-based risk model for estimating the prognostic outcomes of patients suffering from COAD.

**Methods.** Samples were collected from Rectal Adenocarcinoma (TCGA-READ) Pan-Cancer Atlas datasets, The Cancer Genome Atlas Colon Adenocarcinoma (TCGA-COAD), and the GSE87211 dataset. MRNAsi was calculated by one-class logistic regression (OCLR) algorithm. Under the criterion of correlation greater than 0.4, genes related to mRNAsi were screened and clustered. Meanwhile, differentially expressed genes (DEGs) between molecular subtypes were identified to establish a risk model. According to the median risk score value for immunotherapy and results from immune cell infiltration and clinicopathological analyses, clusters and patients were divided into high-RiskScore and low-RiskScore groups. Cell apoptosis and viability were detected by flow cytometer and Cell Counting Kit-8 (CCK-8) assay, respectively.

**Results.** A negative correlation between mRNAsi and clinical stages was observed. Three clusters of patients (C1, C2, and C3) were defined based on a total of 165 survival-related mRNAsi genes. Specifically, C1 patients had greater immune cell infiltration and a poorer prognosis. A 5-mRNAsi-gene signature (HEYL, FSTL3, FABP4, ADAM8, and EBF4) served as a prediction index for COAD prognosis. High-RiskScore patients had a poorer prognosis and higher level of immune cell infiltration. In addition, the five genes in the signature all showed a high expression in COAD cells. Knocking down HEYL promoted COAD cell apoptosis and inhibited viability.

**Conclusion.** Our mRNAsi risk model could better predict the prognosis of COAD patients.

## INTRODUCTION

COAD is a malignant tumor of the digestive tract that mainly derives from the colon gland epithelial cells and is the most common type of colon cancer (*Dienstmann, Salazar & Tabernero, 2015*). According to the World Health Organization (WHO) report in 2018, there are approximately 1.8 million patients suffering from with colon cancer worldwide, and COAD has the fifth highest mortality rate (*Dube et al., 2019*; *Hu et al., 2021b*). Clinical symptoms of COAD at early stage are not obvious, but as the cancer develops, positive fecal octane blood, bloody stool, mucous pus and blood stool, and tenesmus may occur. Patients with at middle and late stage will experience abdominal pain, intestinal obstruction, anemia and some other symptoms (*Li & Martin, 2016*). In clinical practice, patients with COAD are often diagnosed at advanced stage when surgical outcomes are poor or the opportunity for radical surgery is lost. Therefore, accurate genetic markers predictive of COAD survival and prognosis should be developed for clinical diagnosis and treatment of COAD (*Ji, Peng & Wang, 2018*).

Cancer stem cells (CSCs) are a group of cells with multidirectional differentiation and self-renewal potential that maintain tumor heterogeneity and cause tumor growth, recurrence and metastasis (*Li & Li, 2014*). At present, CSCs are considered as a major contributor to the failure of tumor chemoradiotherapy. Therefore, developing targeted drugs against CSCs and further studying the mechanism of drug resistance, metastasis and recurrence caused by CSCs is the key to the development of CSC-targeted therapy (*Li et al., 2021*). In 1994, *Lapidot et al. (1994)* first isolated CSCs with immunophenotype CD34+/CD38 − from acute myeloid leukemia (AML) cells, and they confirmed the self-renewal ability of CSCs in mice with severe combined immunodeficiency. Integration of deep learning method and artificial intelligence helps investigate the features of CSCs (*Malta et al., 2018*). OCLR can quantify stemness by defining signatures and assess oncogenic dedifferentiation. MRNAsi could be calculated based on transcriptome and epigenetic feature sets extracted from untransformed pluripotent stem cells and their differentiated progeny (*Zhang et al., 2020*).

In this study, CSC-related indicators were created based on The Cancer Genome Atlas (TCGA) and the Gene Expression Omnibus (GEO) databases. ConsensusClusterPlus was used to define mRNAsi-based molecular subtypes. Five mRNAsi-related genes were discovered for COAD prognostic risk prediction using univariate regression analysis and least absolute shrinkage and selection operator (LASSO) analyses. Finally, the level of immune cell infiltration in COAD patients from different risk groups was assessed using Estimation of STromal and Immune cells in MAlignant Tumour tissues using Expression data (ESTIMATE) and CIBERSORT algorithms. The 5-mRNAsi-related gene signature could improve the current prognostic prediction for COAD in clinical practice.

## MATERIALS AND METHODS

### Raw data

RNA-seq data of 373 COAD cases with clinical information were acquired from the TCGA-COAD dataset (https://portal.gdc.cancer.gov/), which was used as the training

dataset. The TCGA-READ dataset with 146 COAD cases and the GSE87211 dataset with 197 COAD cases were used as independent validation datasets.

## Calculation of mRNAsi

MRNAsi describes the similarity between stem cells and tumor cells based on gene expression data. The mRNAsi ranges from 0 to 1, with a value close to 1 indicating a less differentiated cells and stronger stem cell properties. In this study, the mRNAsi of cells from the TCGA-COAD datasets was calculated by OCLR algorithm and the stemness model of the Progenitor Cell Biology Consortium (PCBC, https://progenitorcells.org/) (*Malta et al., 2018*; *Wang et al., 2021*).

## Genes associated with mRNAsi

Under the threshold of |cor|>4 and $p < 0.01$, Spearman correlation coefficients and $p$ values were calculated for tumor stemness index of COAD samples and protein-coding genes in TCGA-COAD. Next, survival-associated mRNAsi genes with $p < 0.01$ were filtered using univariate Cox regression analysis by the Coxph function of R package survival.

## Molecular subtypes

Based on prognostic genes related to mRNAsi, molecular subtyping for TCGA-COAD samples was performed in the R package Consensus Cluster Plus 1.52.0 (*Wilkerson & Hayes, 2010*). Km arithmetic and "1-spearman correlation" distance were applied to run 500 bootstraps with each bootstrap involving specimens (≥80%) of TCGA-COAD samples dataset. The optimum k in the range between 2 and 10 was selected per cumulative distribution function (CDF) and consistency matrix.

## Construction of a prognostic model

Firstly, limma package and univariable Cox analysis were employed to screen differentially expressed gene (DEGs) related to both COAD prognosis and mRNAsi among molecular subtypes. LASSO regression in the glmnet package (*Engebretsen & Bohlin, 2019*) was conducted based on the prognosis genes. Finally, a RiskScore formula was developed as follow to evaluate patients' prognosis:

$$\text{Risk Score} = \sum_{k=0}^{n} \beta i \times Expi$$

*Expi* was the expression of the *i* gene and *βi* was the Cox regression coefficient of the *i* gene. Under the optimal threshold (R package survminer), samples in the training dataset and independent validation datasets were classified into low- and high-RiskScore groups. Finally, the "timeROC" package was used to analyze the area under the ROC curve (AUC) for 1-, 3-, and 5-year survival (*Blanche, Dartigues & Jacqmin-Gadda, 2013*). Kaplan–Meier (KM) survival curves between low- and high-RiskScore groups of COAD were generated.

## Gene enrichment and pathway activity analysis

Pathway analysis was conducted using "Fgsea" package in R. All KEGG candidate gene sets were subjected to gene set enrichment analysis. The "ClusterProfiler" package was used for functional annotation (*Yu et al., 2012*).
Furthermore, we aim to explore the oncogenic activity of different clusters in cell-specific signaling pathways such as PI3K, VEGF, EGFR, p53 and MAPK. Here, we do this by using the PROGENy algorithm, which is a method capable of accurately inferring signaling pathway activity from gene expression under different conditions (*Schubert et al., 2018*). The PROGENy algorithm can generate a core set of genes (Pathway RespOnsive GENes) for the corresponding signaling pathway through a large number of publicly available perturbation experiments. Subsequently, this core set of genes (Pathway RespOnsive GENes) and known expression data were used to calculate signaling pathway activity scores, and heat maps were produced to look at pathway activation in different molecular subtypes (*Schubert et al., 2018*).

## Immune cell abundance

The CIBERSORT algorithm (https://cibersort.stanford.edu/) was applied to quantify relative abundance of 22 immune cell types in COAD. Meanwhile, Immune Score, ESTIMATE Score and Stromal Score were calculated by estimation of Stromal and Immune cells in Malignant Tumors using Expression data (ESTIMATE) (*Yang et al., 2021*).

## Drug sensitivity analysis

The half-maximal inhibitory concentration (IC50) values were calculated using the "pRRophetic" package (*Geeleher, Cox & Huang, 2014*) to predict the response to commonly used chemotherapeutic drugs in different risk groups.

## Cell growth and transient transfection

The expression of HEYL, FSTL3, FABP4, ADAM8, and EBF4 in normal colonic epithelial cells NCM460 and colon cancer cell lines HCT116 and SW480 was detected through RT-qPCR assay. NCM460, HCT116 and SW480 cell lines were acquired from Beijing Bena Biotechnology Co. (Beijing, China) and cultured in DEME F-12 medium. Transfection of the negative control (NC), HEYL siRNA (Sagon, China) was performed with Lipofectamine 2000 (Invitrogen, USA). The siRNA sequence targeting HEYL was ATCAACAGTAGCCTTTCTGAATT. The control siRNA sequence (si NC) was AGAAGGCTGGGGCTCATTTG.

## Quantitative reverse transcription-polymerase chain reaction (RT-qPCR)

Total RNA was extracted from NCM460, HCT116 and SW480 cell lines with TRIzol reagent (Thermo Fisher, Waltham, MA, USA). Using a LightCycler 480 PCR System and FastStart Universal SYBR®Green Master (Roche, Indianapolis, IN, USA), RT-qPCR was performed on the the acquired RNA from each sample (2 μg). A reaction volume were prepared in a total amount of 20 μl that consisted of 0.5 μl of forward primer and 10μl reverse primer, required amount of water, and 2 μl of cDNA template, with the cDNA serving as a template. To conduct RT-qPCR reaction, an initial DNA denaturation phase lasted for 30 s (s) at 95 °C, followed by 45 cycles for 15 s at 94 °C, for 30 s at 56 °C, and for 20 s at 72 °C. Each sample was run in triplicates. The data were standardized to the level of GAPDH using the $2^{-\Delta\Delta CT}$ method. See Table 1 for the sequences of primer pairs for the

**Table 1  qRT-PCR primer sequences.**

| Gene | Forward primer sequence (5′–3′) | Reverse primer sequence (5′–3′) |
|------|--------------------------------|--------------------------------|
| HEYL | ATGAAGCGACCCAAGGAGCC | GGCTACTGTTGATGCGGTCT |
| FSTL3 | GTGCCTCCGGCAACATTGA | GCACGAATCTTTGCAGGGA |
| FABP4 | ACTGGGCCAGGAATTTGACG | CTCGTGGAAGTGACGCCTT |
| ADAM8 | GAGGGTGAGCTACGTCCTTG | CAGCCGTATAGGTCTCTGTGT |
| EBF4 | TTCGTGGAAAAGGACCGAGAG | GGCACATTTCGGGGTTCTTG |
| GAPDH | AATGGGCAGCCGTTAGGAAA | GCCCAATACGACCAAATCAGAG |

genes. The experiment was performed following a previous published paper (*Bustin et al., 2009*).

## Flow cytometry

According to the manufacturer's protocols, briefly, trypsin was employed for cell harvesting. Cells at the concentration of $1 \times 10^5/200$ µL were then resuspended in PBS, followed by Annexin V-FITC and PI solution staining on ice for 30 min (min) away from light. The samples were washed using PBS and then analyzed by BD FACS Calibur flow cytometer (BD, Franklin Lakes, NJ, USA).

## Cell counting Kit-8 assay (CCK-8)

CCK-8 assay (Beyotime, Jiangsu, China) was conducted strictly following the protocol. Cells with different treatments were cultured at a density of $1 \times 10^3$ cells/well in 96-well plates and added with CCK-8 solution. A microplate reader (Thermo Fisher, USA) was applied to detect the OD 450 values of each well were detected using after 2-hour (h) incubation at 37 °C.

# RESULTS

## Predicting overall survival for COAD patients by mRNAsi

The correlation analysis showed that mRNAsi was correlated with the N Stage, T stage, and Stage (Fig. 1A). Samples in the TCGA-COAD dataset was divided into the high-mRNAsi group and low-mRNAsi group, where the high mRNAsi group demonstrated a better survival outcome (Fig. 1B). Moreover, samples with late clinical features had a low mRNAsi (Fig. 1C). The survival outcomes of mRNAsi-high and mRNAsi-low of Stage I–IV samples indicated that Stage IV samples with higher mRNAsi had shorter survival time (Fig. 1D).

## Identification of three clusters based on mRNAsi

A total of 165 mRNAsi-related genes of prognostic significance were screened using spearman analysis and univariate Cox regression analysis. According to CDF (Figs. 2A) and delta area (2B), samples in TCGA-COAD were classified three clusters when $k = 3$ (Fig. 2C). KM survival curve showed that C3 patients with higher mRNAsi survived longer, while C1with lower mRNAsi had a worse survival outcome (Figs. 2D, 2E). The expression heatmap of 165 mRNAsi-related genes in the three clusters showed that the expression

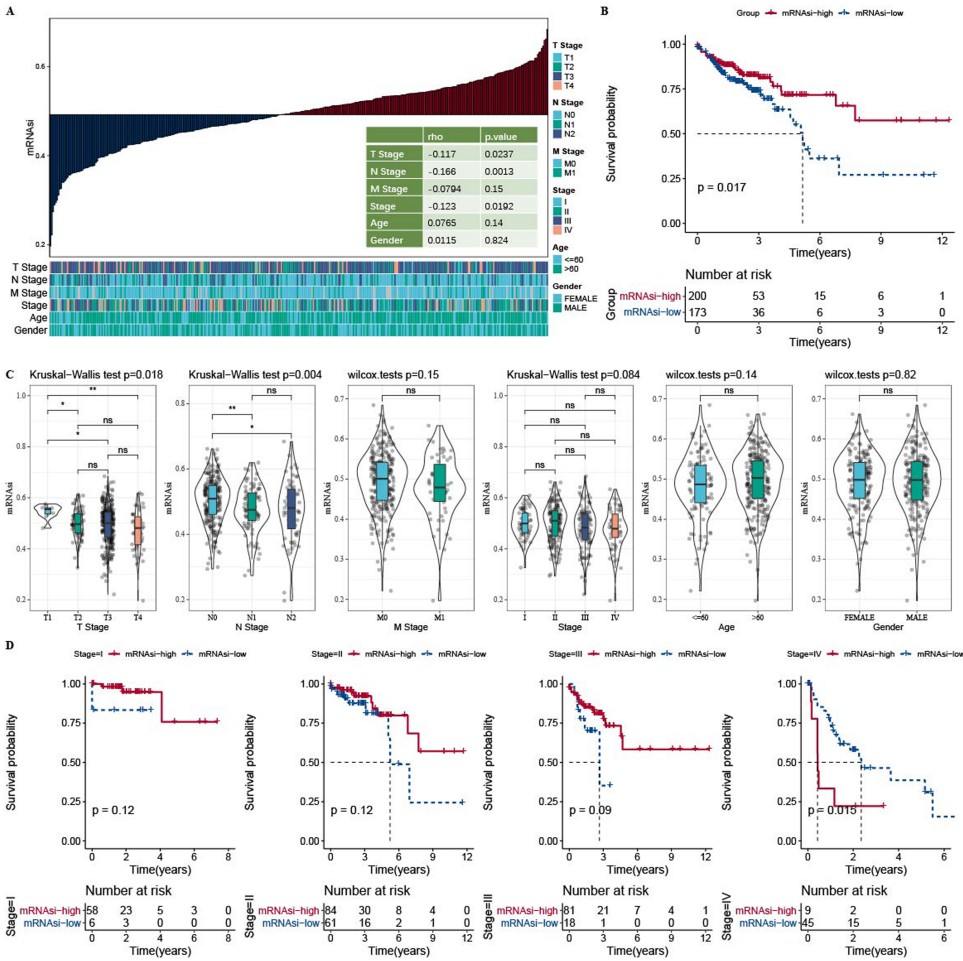

**Figure 1** **Higher mRNAsi predicted prolonged overall survival in Patients with COAD.** (A) Correlation analysis between clinicopathological features and mRNAsi in TCGA-COAD cohort. (B) High and low mRNAsi groups were analyzed by survival analysis. (C) mRNAsi difference analysis among various clinicopathological features. (D) Analysis of survival differences in the two mRNAsi groups by Stage I-IV.

of risk genes was higher in C1, while protective genes showed a higher expression in C3 (Fig. 2F). Remarkable differences in Stage, T Stage, mRNAsi, Status N Stage in the 3 clusters of TCGA-COAD cohort study were observed (Figs. 3A–3H).

## Genomic landscape among molecular subtypes

Molecular characteristic information and published molecular subtypes were acquired from previous research (*Thorsson et al., 2018*). C1 had higher intratumor heterogeneity, loss of heterozygosity, homologous recombination defects, while C3 had higher purity (Fig. 4A). Published molecular subtypes included CIN, GS, HM indel, and HM SNV, and most samples in C1 were CIN (Fig. 4B). ARID1A, PTPRS and KIF26B genes showed extensive somatic mutations in COAD (Fig. 4C).

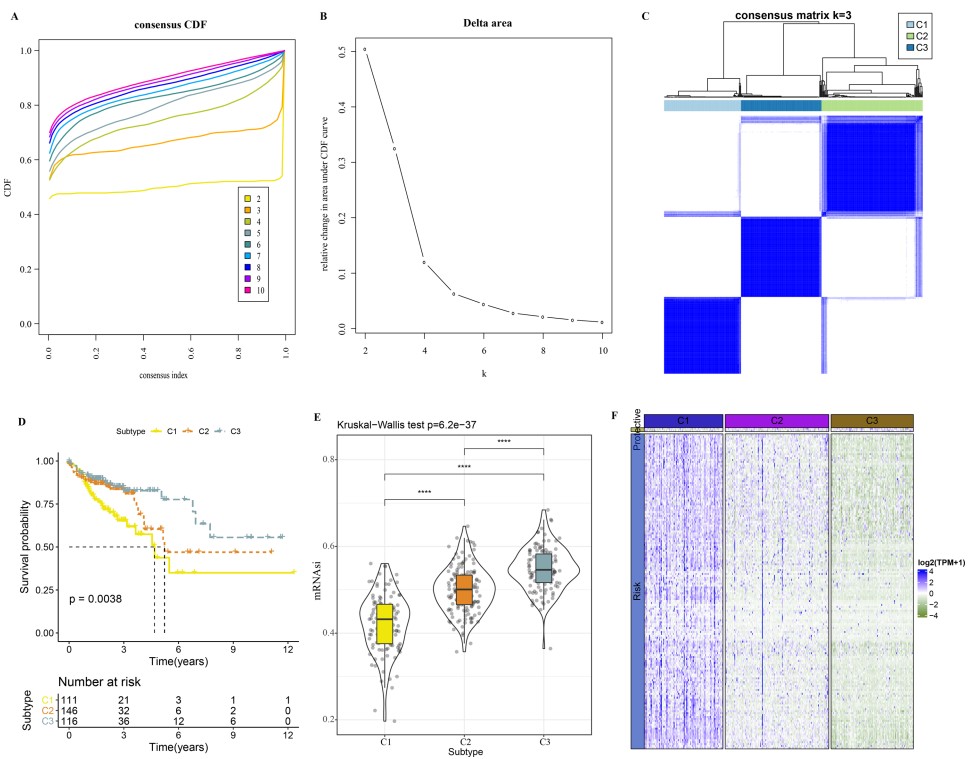

**Figure 2** **Identification of 3 clusters.** (A) Cumulative distribution function. (B) Delta area of CDF. (C) Sample clustering heatmap when $k = 3$. (D) KM survival curve of three clusters. (E) Differences in mRNAsi among three clusters in the TCGA-COAD cohort. (F) Heatmap of mRNAsi-related genes with prognosis in TCGA-COAD dataset.

## Immune infiltration analysis in the 3 clusters

In TCGA-COAD cohort, dendritic_cells_activated, T_cells_CD4_memory_activated, Plasma_cells, T_cells_CD4_memory_resting were enriched in C3, while Macrophages_M1 and Macrophages_M2, Macrophages_M0 were enriched in C1 (Fig. 5A). C1 had higher ImmuneScore, ESTIMATEScore, and StromalScore, as shown by the results of ESTIMATE analysis (Fig. 5B).

In addition, we obtained 29 gene signatures from previous studies (*Bagaev et al., 2021*). Among these 29 gene signatures, ssGSEA results showed that Angiogenesis, fibroblasts, Pro tumor Immune infiltrate, EMT signature were enriched in C1 (Figs. 5C, 5D).

PROGENy algorithm (Pathway RespOnsive GENes) was used to calculate the oncogenic activity of cell-specific signaling pathways, and we observed that JAK-stat, NF-kB, TNF-a, TGF-b, p53, MAPK, Hypoxia pathways were activated in C1 (Figs. 5E, 5F).

## Immunotherapy analysis in the three clusters

Firstly, T cell-inflamed GEP score was used to predict potential in cancer immunotherapy, Th1/IFNγ gene signature (a cytokine with a key function in immune regulation and anticancer immunity) and CYT score to reflect cytotoxic effect were higher in C1

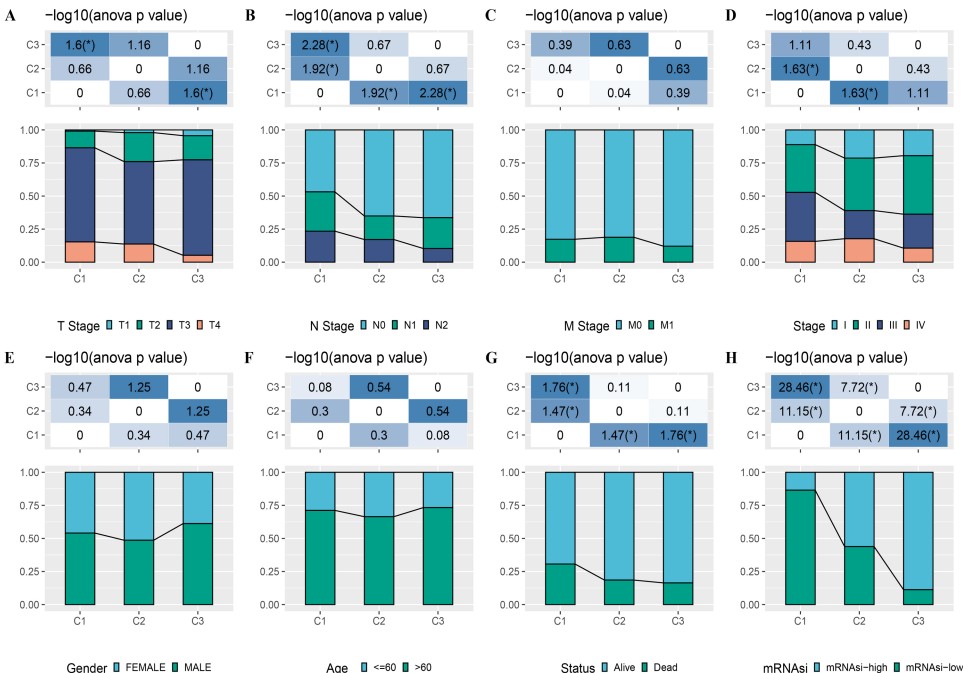

**Figure 3** Distribution of clinical features, including T Stage (A), N Stage (B), M Stage (C), Stage (D), Gender (E), Age (F), Status (G) and mRNAsi (H) in three clusters.

(Figs. 6A–6C). C1 showed upregulated expression of eight immune checkpoint genes (Fig. 6D).

In addition, we collected therapeutic signatures from a previous study, which included gene signatures predicting radiotherapy, oncogenic pathways, and gene signatures related to targeted therapy (*Hu et al., 2021a*). These signatures have the potential to shape the non-inflamed tumor microenvironment (TME). The enrichment score of those gene signatures was calculated by the ssGSEA method. These therapeutic signatures showed significant differences in the enrichment score among the three clusters (Fig. 6E).

## Identification of DEGs among the 3 clusters

Limma analysis identified a total of 612 DEGs including nine differentially downregulated genes and 603 differentially upregulated genes in C1 (Fig. 7A), 500 differentially upregulated genes in C3 (Fig. 7B). GO and KEGG analysis demonstrated that differentially upregulated genes in C1 were enriched in EMT-related pathways (Fig. 7C), differentially downregulated genes in C3 were also enriched in EMT-related pathways (Fig. 7D).

## Establishment of a RiskScore model for predicting COAD prognosis

Based on our identification of the three molecular subtypes, we screened a total of 656 DEGs among the three subtypes using the limma package. Subsequently, 110 prognosis-associated genes were further identified by univariate Cox analysis, and 20 genes with the most significant correlation with mRNAsi were determined. LASSO analysis in glmnet package showed trajectory and confidence interval of lambda

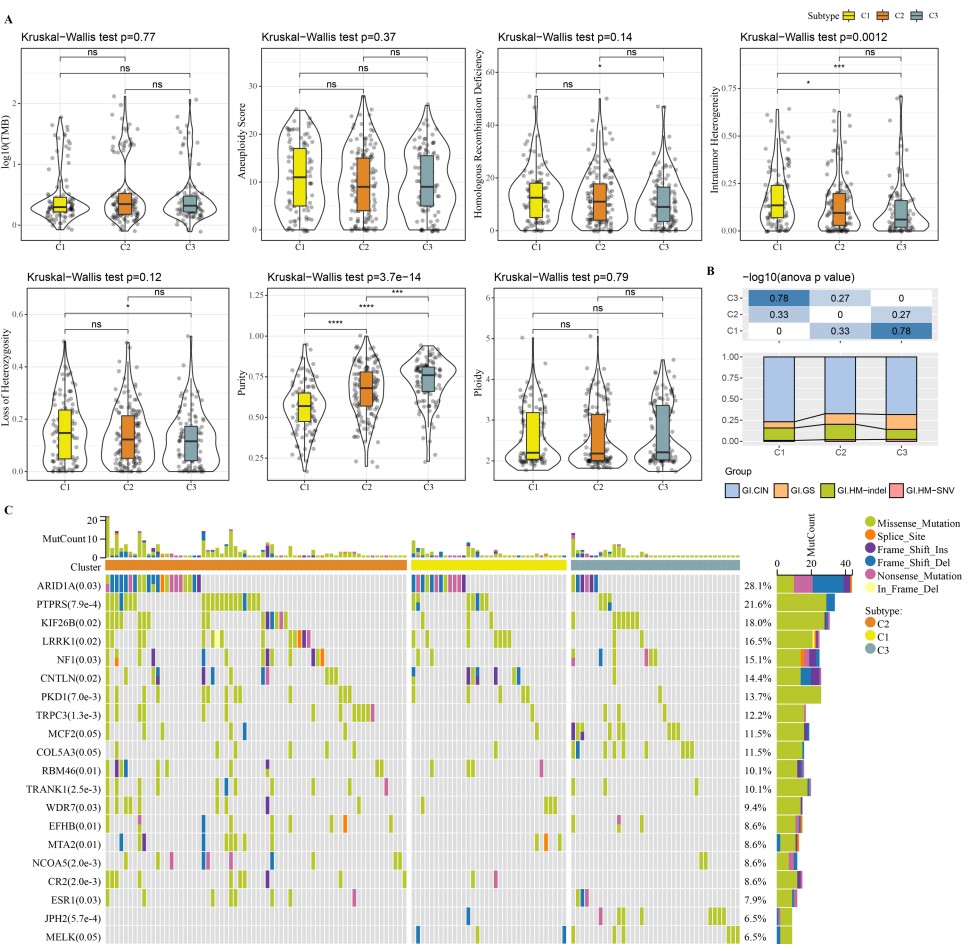

**Figure 4  Genome Landscape.** (A) The differences of number of segments, fraction altered, aneuploidy score, tumor mutation burden, homologous recombination defects, of clusters in the TCGA-COAD dataset. (B) Distribution of published molecular subtypes in three clusters. (C) Somatic mutation in three clusters.

(Figs. S1A–S1B). When lambda = 0.0244, 5 genes were used as model genes: RiskScore = 0.278*HEYL+0.091*FSTL3+0.07*FABP4+0.115*ADAM8+0.057*EBF4

Next, the RiskScore was calculated for patients in TCGA-COAD dataset, and we observed a shorter survival of samples in high-RiskScore group (Fig. 8A). High- and low-RiskScore groups of patients were divided by the cutoff, and KM survival curves showed an overall better survival low-RiskScore group from the TCGA-COAD dataset (Fig. 8B). In the TCGA-COAD cohort, AUC for 1-year, 3-year and 5-year survival was 0.63, 0.64 and 0.72, respectively (Fig. 8C). Low-risk samples in TCGA-READ cohort also survived longer than those with high risk (Fig. 8D), with an AUC of 0.74, 0.75 and 0.66 for 1-year, 3-year and 5-year survival, respectively (Fig. 8E). In GSE87211 cohort, samples in low-RiskScore survived longer (Fig. 8F), with an AUC of 0.73, 0.67 and 0.64 for 1-year, 3-year and 5-year survival, respectively (Fig. 8G).

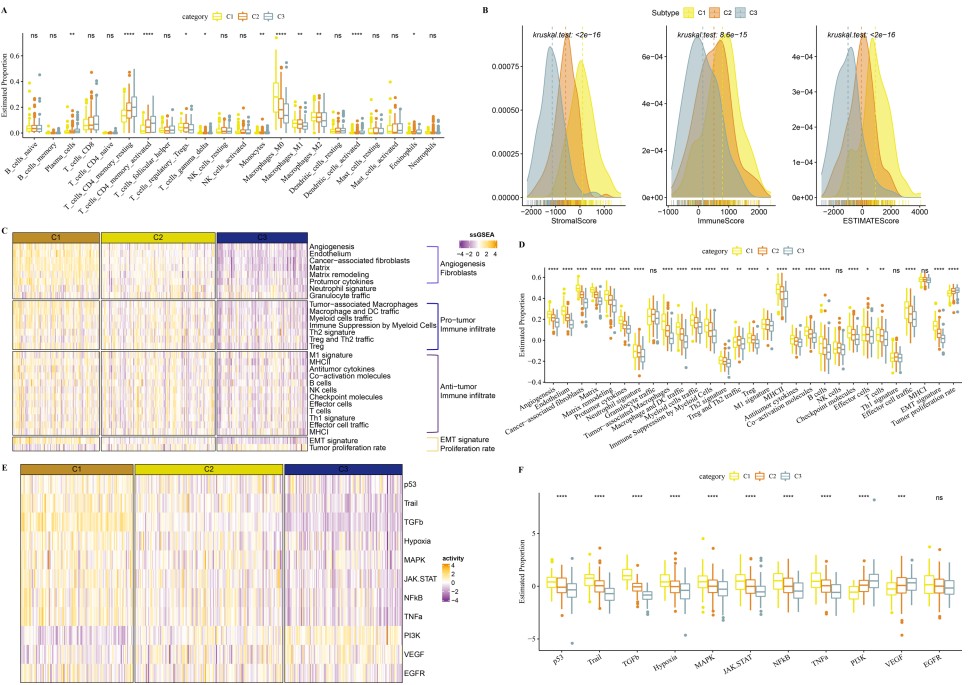

**Figure 5 Immune infiltration analysis in three clusters.** (A) The differences of 22 kinds of immunes score in three clusters. (B) The differences of StromalScore, ImmuneScore and ESTIMATEScore in three clusters. (C) Distribution heatmap of 29 TME-associated gene signatures in three clusters of TCGA-COAD cohort. (D) Statistical difference of 29 TME-associated gene signatures in three clusters of TCGA-COAD cohort. (E) PROGENy was used to measure relative signaling pathway activity scores in tumor cells in TCGA-COAD. (F) Differences in relative signaling pathway activity scores in tumor cells were shown in the box plots.

## Inhibiting the expressions of HEYL promoted the apoptosis of colon cancer cells

We observed elevated expression levels of HEYL, FSTL3, FABP4, ADAM8, and EBF4 in HCT116 and SW480 cell lines in comparison to normal colonic epithelial cells NCM460 (Figs. 9A–9E). This was in line with the predictions of the risk score model. Subsequently, the expression of HEYL in HCT116 and SW480 cell lines was suppressed applying siRNA, and we observed that the proportion of apoptotic HCT116 and SW480 cells (Figs. 9F–9G) was increased and cell viability was reduced (Figs. 9H–9I).

## The distribution of RiskScore in clinical features

The RiskScore difference analysis showed that higher clinical grade had higher RiskScore, and that samples in low-mRNAsi group and C1 had higher RiskScore (Fig. 10A). The majority samples in high-RiskScore were C1 patients and low-mRNAsi patients (Fig. 10B). Moreover, patients in Stage I+II, Stage III+IV, mRNAsi-high, mRNAsi-low, C1, C2 and C3 with a low RiskScore had better survival (Fig. 10C).

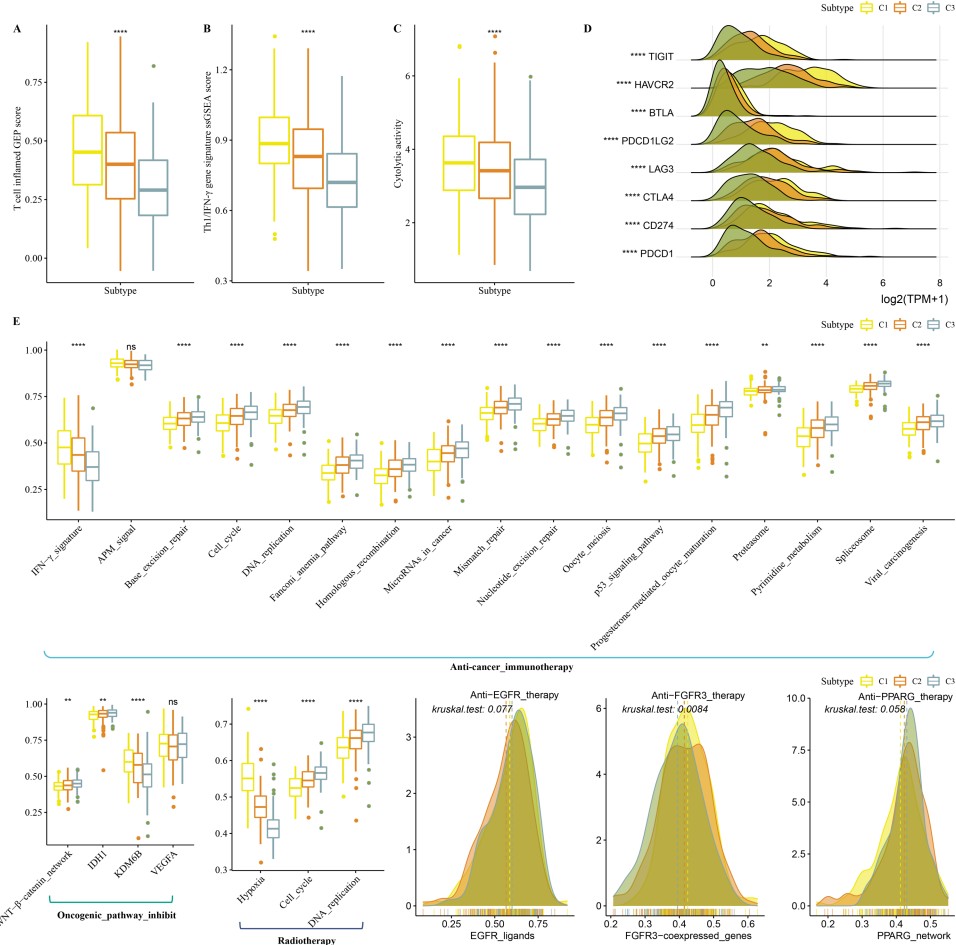

**Figure 6** **Immunotherapy analysis in three clusters.** (A) The differences of T cell inflamed GEP score in three clusters. (B) The differences of Th1/IFN$\gamma$ gene signature in three clusters. (C) The differences of Cytolytic activity in three clusters. (D) The expression level of immune checkpoint genes in three clusters. (E) Enrichment score differences of therapeutic signatures in three clusters.

## Immune infiltration analysis between the two risk groups

GSEA analysis showed that high-RiskScore samples were enriched in CAF and EMT-related pathways (Fig. 11A). There was negative correlation between RiskScore and mRNAsi (Fig. 11B). CIBERSORT analysis showed that T_cells_CD4_memory_resting, Plasma_cells, and T_cells_CD4_memory_activated were enriched in RiskScore-low group (Fig. 11C). 22 immune cells were correlated with each other (Fig. 11D). ESTIMATE analysis showed that StromalScore, ImmuneScore and ESTIMATEScore were higher in the high-RiskScore group (Fig. 11E). RiskScore was positively related to fibroblasts, pro-tumor immune infiltration, angiogenesis, and EMT-related gene signatures (Fig. 11F). The pathway and RiskScore analysis demonstrated a positive correlation between the RiskScore and pathways related to angiogenesis (Fig. 10G).

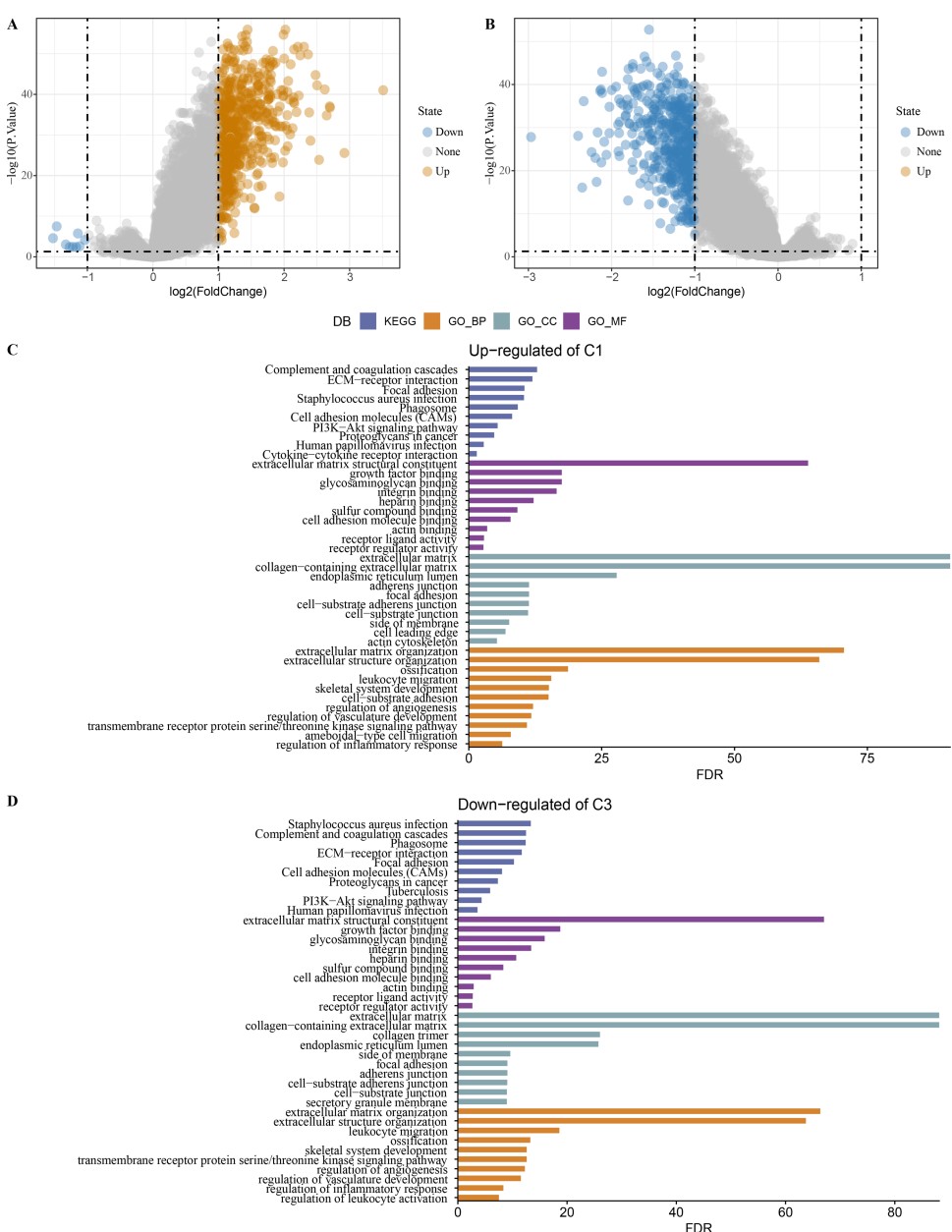

**Figure 7  Identification of differentially expressed gene.** (A) Differentially expressed genes in C1 *vs* other of the TCGA-COAD dataset were shown in the volcano diagram. (B) Differentially expressed genes in C3 *vs* other of the TCGA-COAD dataset were shown in the volcano diagram. (C) Differentially upregulated genes in C1 *vs* other were analyzed by functional enrichment analysis. (D) Differentially downregulated genes in C3 *vs* other were analyzed by functional enrichment analysis.

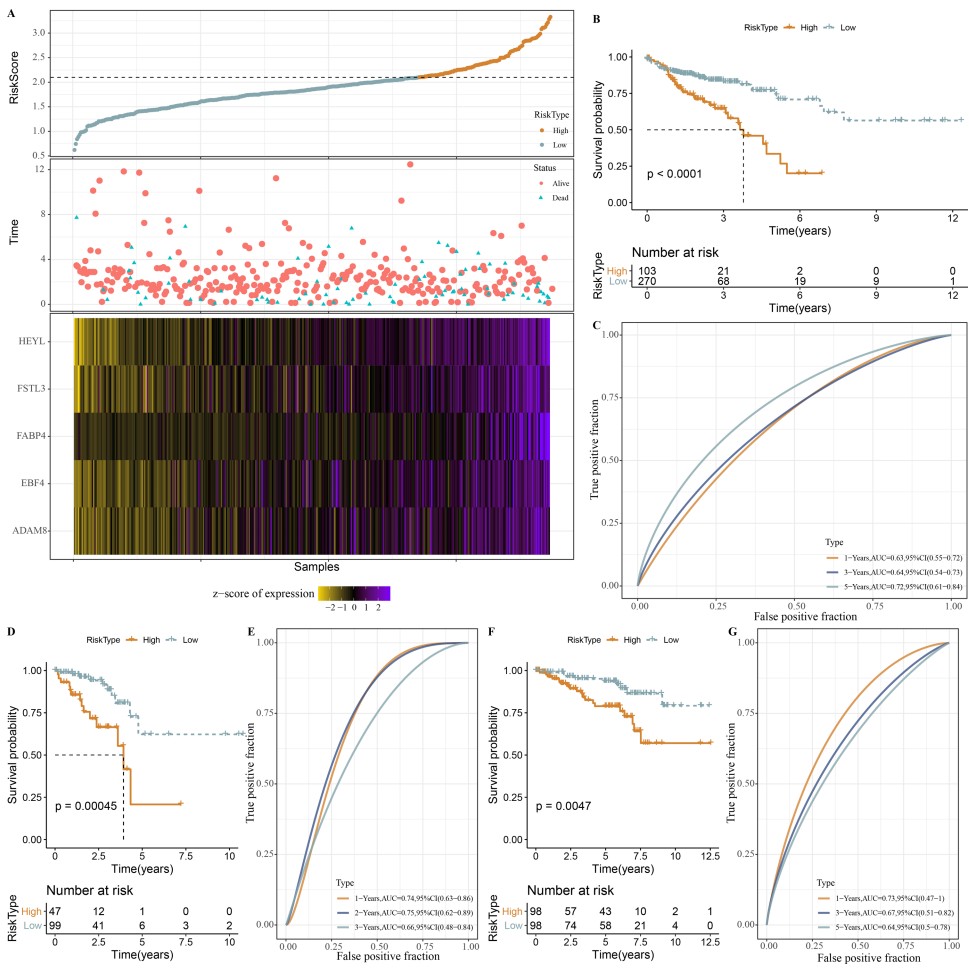

**Figure 8  Establishment and validation of RiskScore.** (A) RiskScore, survival time, status, mRNAsi-related prognosis genes expressions in TCGA-COAD dataset. (B) ROC KM survival curve of high RiskScore group and low RiskScore group in TCGA-COAD dataset. (C) ROC analysis of RiskScore in TCGA-COAD dataset. (D) KM survival curve of high RiskScore group and low RiskScore group in TCGA-READ dataset. (E) ROC analysis of RiskScore in TCGA-READ dataset. (F) KM survival curve of high RiskScore group and low RiskScore group in GSE87211 dataset. (G) ROC analysis of RiskScore in GSE87211 dataset.

## Immunotherapy analysis between high-RiskScore and low- RiskScore groups

Tumor immunotherapy is thought to be effective in treating a variety of tumors (*Cui, Peng & Chen, 2022*). In this study, the high- RiskScore group showed higher Th1/IFNγ gene signature, T cell inflamed GEP score, and CYT score (Figs. 12A-12C). The expression of eight immune checkpoint genes was increased in high- RiskScore group (Fig. 12D). Furthermore, RiskScore was positively correlated with Th1/IFNγ gene signature, T cell-inflamed GEP score, CYT score and immune checkpoint genes (Fig. 12E). Susceptible diversity of common chemo medicines between the two groups was investigated and the

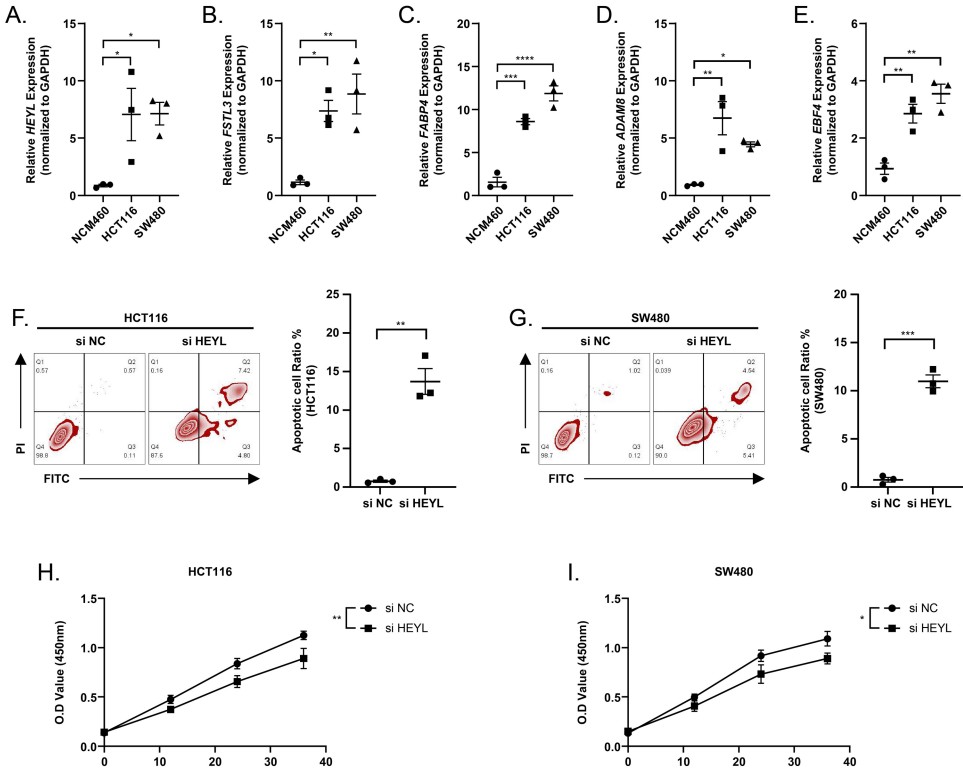

**Figure 9** **Assessing the reliability of a risk score model through experimental testing.** (A–E) The expression of HEYL, FSTL3, FABP4, ADAM8, and EBF4 in NCM460, HCT116 and SW480 cells ($n = 3$) was detected by PCR. (F–G) Representative apoptosis results after inhibition of HEYL expression in HCT116 and SW480 cell lines ($n = 3$). (H-I) Representative CCK8 results after inhibition of HEYL expression in HCT116 and SW480 cell lines ($n = 3$). $* \leq 0.05, ** \leq 0.01, *** \leq 0.001, **** \leq 0.0001$. The results are presented as mean $\pm$ S.E.M.

results indicated that the IC50 of gefitinib, vinorelbine, and cisplatin were higher in high-RiskScore group and 5-Fluorouracil was higher in low-RiskScore group (Fig. 12F).

## DISCUSSION

Study reported that conventional therapies do not target CSCs, therefore tumors can eventually be regenerated by surviving CSCs (*Cherciu et al., 2014*). The interaction between stromal and cancerous cells in solid tumors is mainly responsible for non-cell-autonomous resistance and could extensively promote tumor growth in tumor immune microenvironment. Cell-autonomous resistance is also known as therapeutic resistance characteristics of cancer cells, especially for CSCs (*Qu et al., 2019*).

This study analyzed the correlation of mRNA index in COAD tissues based on OCLR (*Malta et al., 2018*). High-mRNAsi samples were found to have a better prognosis. By comparing mRNAsi with clinical features, we found that patients with higher Stage also had significantly higher mRNAsi. The results suggested the potential correlation between Stage and CSCs. Furthermore, three molecular subtypes (C1, C2, and C3) showing significant differences in immune characteristics, immunotherapy sensitivity and prognosis were

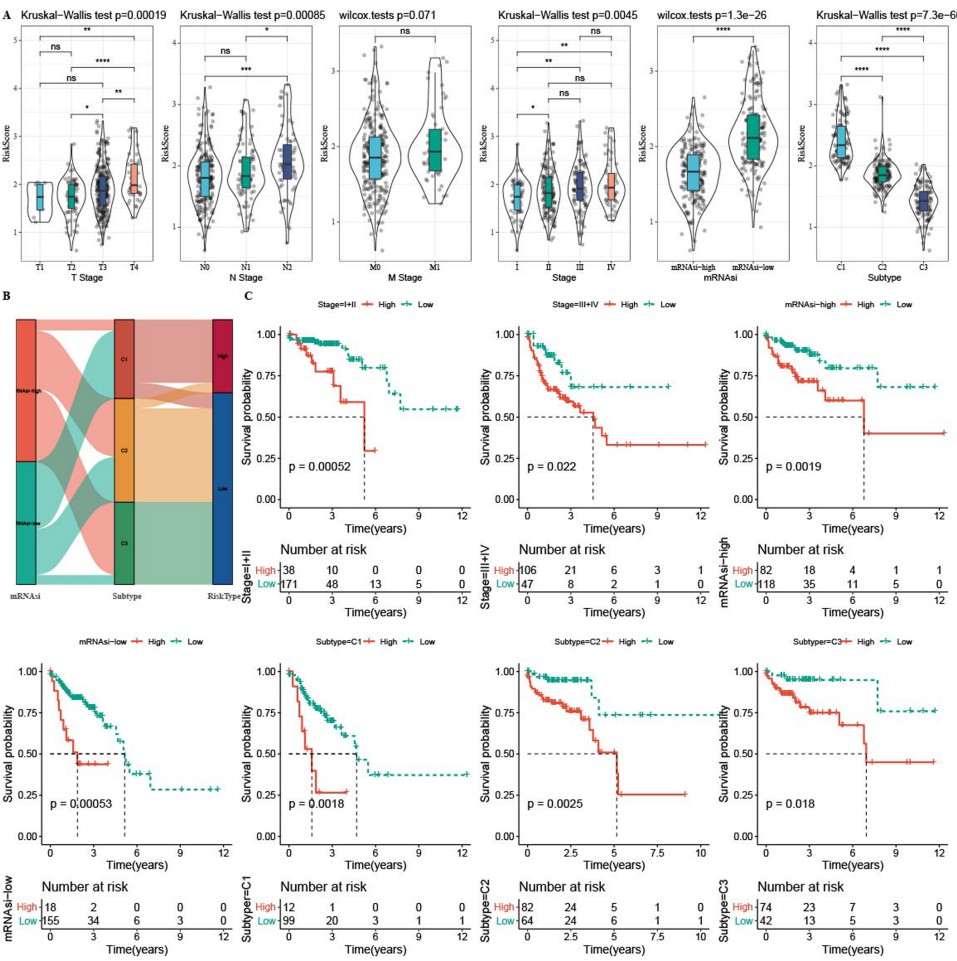

**Figure 10** **The distribution of RiskScore in clinical features.** In the TCGA-COAD cohort: (A) differences in the distribution of RiskScore among different clinicopathological groups. (B) distribution of RiskScore in published molecular subtypes and mRNAsi. (C) KM curves between high and low risk groups among different clinicopathological groups.

defined based on mRNAsi. Notably, our study demonstrated that C1 subtype had higher intratumor heterogeneity, loss of heterozygosity, and homologous recombination defects, and that the majority of C1 patients had low mRNAsi, indicating that mRNAsi-related genes were responsible for a poorer prognosis in C1. These data demonstrated that the mRNAsi may be used as an indicator to distinguish the heterogeneity within COAD and help decide suitable clinical treatment choice for COAD patients.

Key genes were screened for COAD and further analyzed for their effects on the cancer using functional annotation. The main biological processes were related to the regulation of the PI3K-Akt signaling pathway, cell cycle, EMT-related pathways, ECM receptor interaction, MAPK signaling pathway, which all have been reported to promote self-renewal, apoptosis inhibition, and cell survival of CSCs.

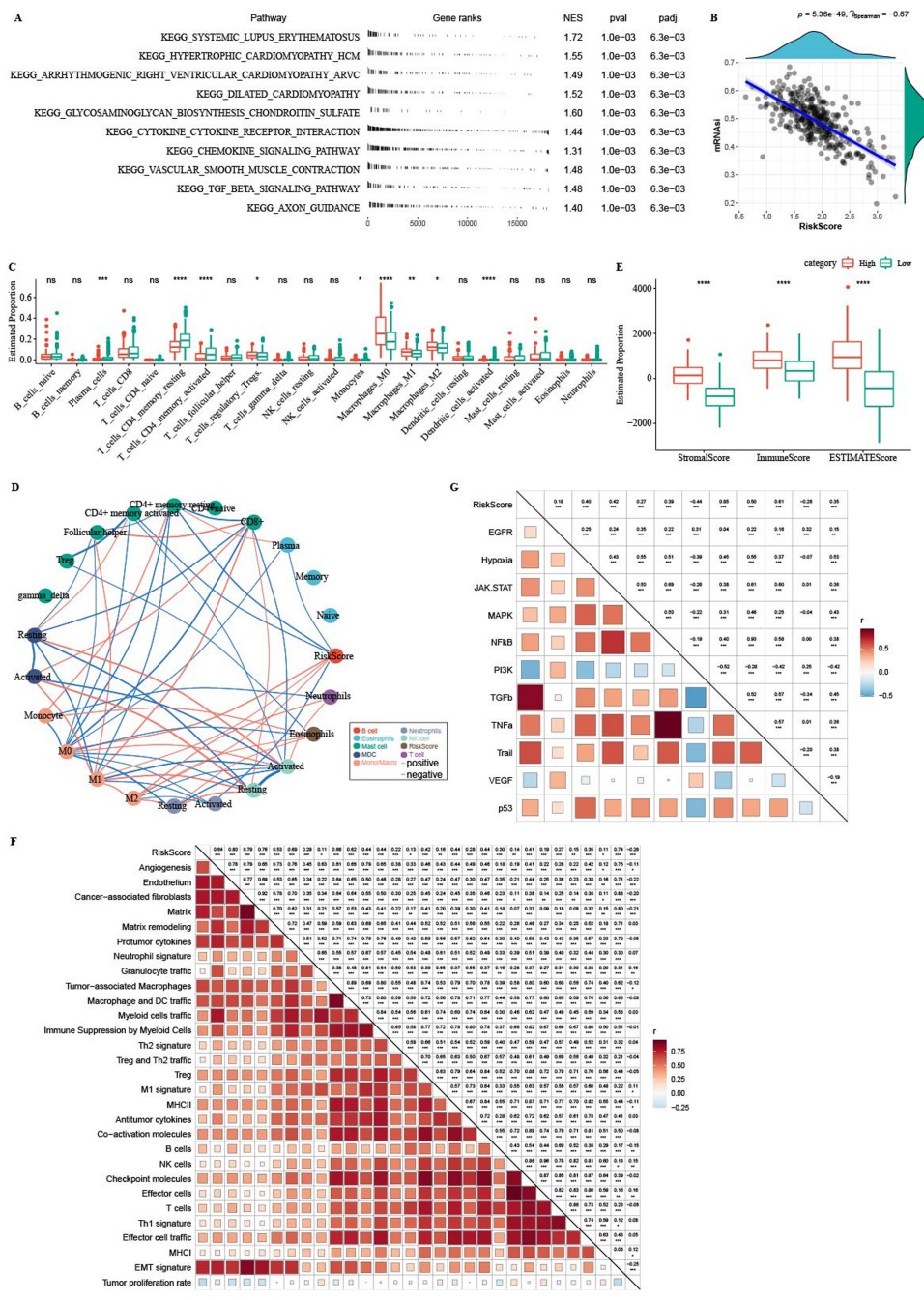

**Figure 11 Immune microenvironment analysis.** (A) The fgsea algorithm was performed with all KEGG gene sets in the high RiskScore and low RiskScore groups. (B) RiskScore. There was negatively correlated between mRNAsi and RiskScore. (C) The differences of 22 kinds immunes score between high group and low group. (D) The correlation analysis of 22 kinds immune cells. (E) The differences of StromalScore, ImmuneScore and ESTIMATEScore between high group and low group. (F) The correlation between 22 immune cells scores and RiskScore. (G) The correlation between pathways scores and RiskScore.

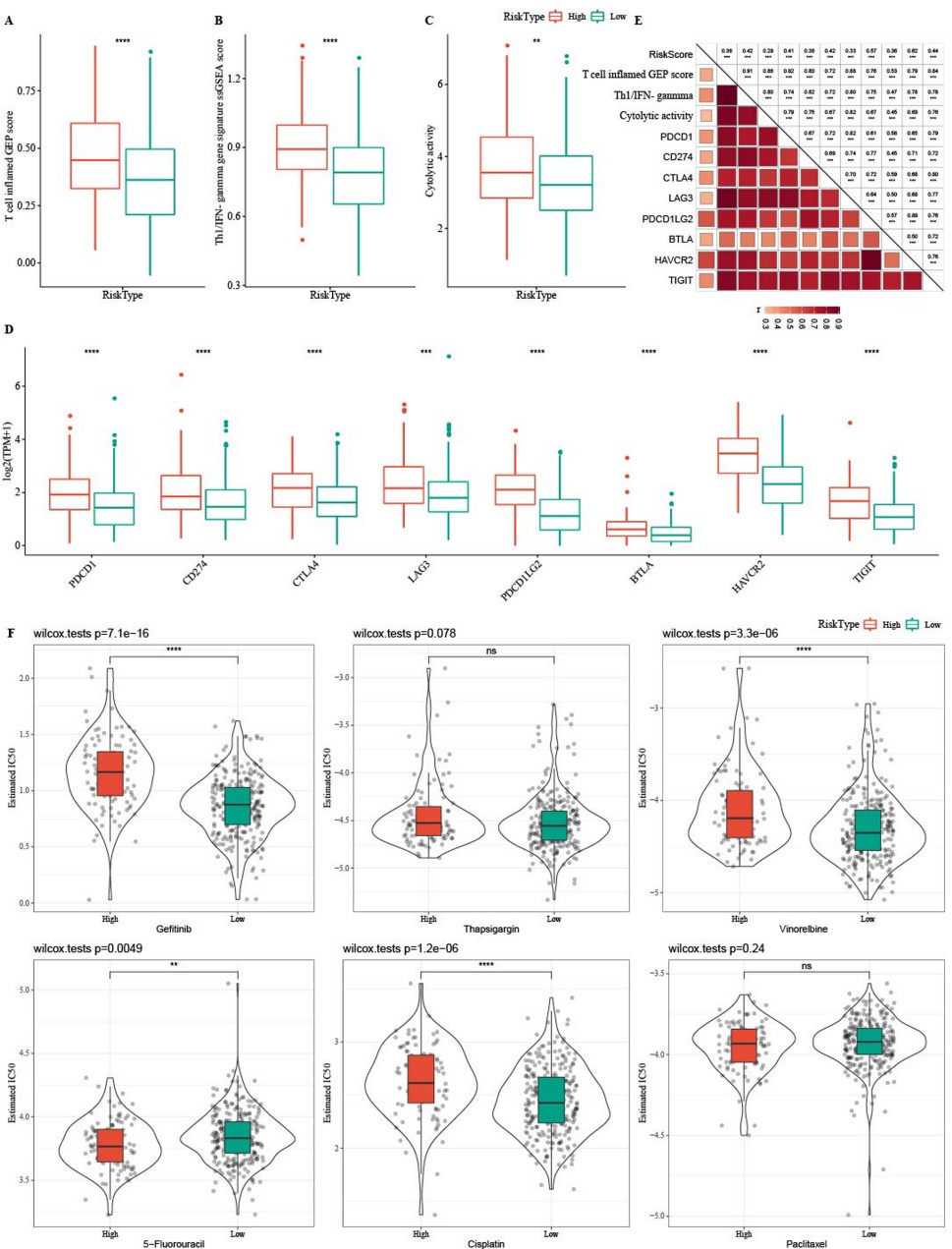

**Figure 12** **Immunotherapy analysis.** (A) The differences of T cell inflamed GEP score in the high RiskScore and low RiskScore groups. (B) The differences of Th1/IFNγ gene in the high RiskScore and low RiskScore groups. (C) The differences of Cytolytic in the high RiskScore and low RiskScore groups. (D) The differences expressions of immune checkpoint genes in the high RiskScore and low RiskScore groups. (E) Correlation analysis between RiskScore and immune checkpoint gene expression, T cell inflamed GEP score, Cytolytic activity, and Th1/IFNγ gene signature. (F) The box plots of the estimated IC50 for gefitinib, thapsigargin, vinorelbine, 5-fluorouracil, cisplatin and paclitaxel in TCGA-COAD.

*Pan et al. (2019)* identified 13 key genes for bladder cancer based on mRNAsi-related genes. In colorectal cancer, analysis on the impact of RCN3 on patients' prognosis showed that the expression level of RCN3 is an independent risk factor for a poor prognosis and response to chemotherapy (*Ma et al., 2022*). These studies showed that the expression of key genes is closely related to the OS of patients. In this study, five mRNAsi-related genes were identified as independent predictors for the OS of COAD patients and further utilized to generate a RiskScore for COAD. Based on various clinical features required for the prognostic models, COAD samples were easily classified into high-RiskScore and low-RiskScore groups by the mRNAsi signature. This supported a potential clinical application significance of the mRNAsi signature.

Our mRNAsi signature consisted of HEYL, FSTL3, FABP4, ADAM8 and EBF4. HEYL as an important downstream effector of Notch pathway is high-expressed in some tumors related to estrogen (*Leimeister et al., 2000*). HEYL-aromatase axis promotes CSCs *via* endogenous estrogen-induced autophagy in castration-resistant prostate cancer (*Lin et al., 2021*). Transformation of pluripotent stem cells to cardiogenic endothelial cells could be increased by FSTL3 (*Kelaini et al., 2018*), but its role is unclear in tumor stem cells. In addition, *in vitro* experiments confirmed that the five genes were low-expressed in COAD cell lines, and that knocking out HEYL promoted apoptosis and inhibited cell viability. These phenomena indicated that those genes may be vital for progression and occurrence of COAD. Using enzyme-linked immunosorbent assay, *Ru et al. (2019)* detected that the level of FABP4 is significantly higher in patients' serum than that before surgery, and they suggested that FABP4 expression is related to the pathogenesis of colorectal cancer. ADAM8 is expressed as an antigen of tyrosine kinase inhibitor-resistant chronic myeloid leukemia cells in a model of chronic myeloid leukemia stem cells produced from chronic myeloid leukemia-induced pluripotent stem cells (*Miyauchi et al., 2018*). In addition, *Liu et al. (2022)* identified ADAM8 as one of the key genes associated with metastasis and recurrence in colorectal cancer by WGCNA. In particular, ADAM8 could promote colon cancer cell invasion by activating the TGF-β/Smad2/3 signaling pathway to induce EMT. Similarly, in our study we found that both differentially upregulated genes in C1 and differentially downregulated genes in C3 were enriched in EMT-related pathways. The early B cytokine (EBF) transcription factor family member EBF4 enhances cytotoxic activity in human immune cells and affects FAS-mediated apoptosis (*Kubo et al., 2022*).

Potential mRNAsi to estimate COAD prognosis from large samples has been identified applying bioinformatics analysis, but the current study still had some limitations. Firstly, our samples lacked clinical follow-up information, which excluded potential factors such as patients' health conditions when screening biomarkers. Also, the results were not convincing enough as they were obtained only by bioinformatics analyses, which therefore required further experimental verification. Finally, the molecular processes and signaling pathways obtained only from TCGA cases should be confirmed in other datasets in the future.

In summary, we developed an mRNAsi-based gene signature that can accurately prognosis, stage, mutation, immune profile, and immunotherapeutic differences for COAD patients, providing a reliable guideline for further studies on the mechanisms

of COAD heterogeneity. In addition, a prognostic system was produced based on five mRNAsi-related genes as a prognostic indicator. In particular, the role of HEYL in COAD stem cell characterization will be explored in depth in further studies. The current classifier may be utilized as an accurate molecular diagnostic tool for assessing the prognostic risk of COAD patients.

### Funding

The authors received no funding for this work.

### Competing Interests

The authors declare there are no competing interests.

### Author Contributions

- Haifu Huang conceived and designed the experiments, analyzed the data, authored or reviewed drafts of the article, and approved the final draft.
- Lin Lu conceived and designed the experiments, analyzed the data, authored or reviewed drafts of the article, and approved the final draft.
- Yaoxuan Li conceived and designed the experiments, analyzed the data, authored or reviewed drafts of the article, and approved the final draft.
- Xiumei Chen conceived and designed the experiments, performed the experiments, analyzed the data, prepared figures and/or tables, authored or reviewed drafts of the article, and approved the final draft.
- Meng Li performed the experiments, prepared figures and/or tables, and approved the final draft.
- Meiling Yang performed the experiments, prepared figures and/or tables, and approved the final draft.
- Xuewu Huang performed the experiments, prepared figures and/or tables, and approved the final draft.

### Data Availability

The data is available at NCBI GEO: GSE87211.

The data analyzed in this study is available at Github and Zenodo:

– https://github.com/xuewuhu/Raw-data.git

– xuewuhu. (2023). xuewuhu/Raw-data: First release of my data (v1.0.0). Zenodo. https://doi.org/10.5281/zenodo.8417835

### Supplemental Information

Supplemental information for this article can be found online at http://dx.doi.org/10.7717/peerj.16477#supplemental-information.

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
