# Peer review of "Development of a 5-mRNAsi-related gene signature to predict the prognosis of colon adenocarcinoma"

_PeerJ, doi:10.7717/peerj.16477_

## Round 0.1 · original submission · Major Revisions

Please respond and make appropriate revisions based on the reviewers' suggestions and my comments (below). This will greatly improve the quality of the manuscript.

My comments:
1. Lines 88-93: The final lines summarize the study's goals but could benefit from more specificity. What are the expected outcomes or the novelty in the approach? How does the study intend to perform immune function analysis, and why is it relevant to COAD prognosis?
2. "which was acted as training dataset" could be revised to "which was used as the training dataset.
3. Line 152: In the description of the RT-qPCR method, there seems to be a typo or missing unit in the description of RNA ("2 g"). Ensuring that all units and concentrations are accurate and consistent would enhance the methodological clarity.
4. Line 147: In the description of the RT-qPCR method, there seems to be a typo or missing unit in the description of RNA ("2 g"). Ensuring that all units and concentrations are accurate and consistent would enhance the methodological clarity.
5. "which high mRNAsi group had better survival outcome" could be revised to "where the high mRNAsi group demonstrated a better survival outcome."
6. Line 173: The first sentence mentions that higher mRNAsi predicted prolonged overall survival, but later mentions that Stage IV samples with higher mRNAsi had short survival time? Clarifying the conditions and contexts for these contrasting statements would help in understanding.
7. Line 204: There are minor typographical errors that need correction, such as "endritic_cells_activated" (probably should be "dendritic_cells_activated").
8. The term "Stage Stage" appears in the sentence "potential correlation between Stage Stage and CSCs." This seems like a typo and should be corrected to just "Stage".
9. The sentence "Adipocyte differentiation of human adipose-derived stem cells could by inhibited by LncRNA MIR31HG" seems to have a typo. It should likely be "could be inhibited."

**Language Note:** The review process has identified that the English language must be improved. PeerJ can provide language editing services - please contact us at copyediting@peerj.com for pricing (be sure to provide your manuscript number and title). Alternatively, you should make your own arrangements to improve the language quality and provide details in your response letter. – PeerJ Staff

Reviewer 1 ·

Basic reporting

1. While the abstract is informative, its length could be reduced without sacrificing clarity. Condense the abstract to highlight the study's key objectives, methods, findings, and significance.
2. The introduction introduces COAD and CSCs, but it lacks a clear transition to the study's aim. Enhance the transition to better guide the reader toward the study's purpose.

Experimental design

no comment

Validity of the findings

1. Revise the aim section to provide a succinct statement of the study's specific objectives, emphasizing the development and validation of the mRNAsi-based prognostic model.
2. While the methods are outlined, some of the technical details could be streamlined. Focus on explaining the essential steps of the mRNAsi calculation, gene selection, and the development of the prognostic model.
3. Consider reorganizing the results section by grouping related findings together. This would facilitate easier comparison and interpretation of the data.

Additional comments

1. Strengthen the integration of the discussion with the results. For each key finding, include a brief interpretation in the results section and a more extensive discussion in the subsequent section.
2. Elaborate on the clinical implications of the three identified mRNAsi clusters. Discuss how these clusters may guide treatment decisions or provide insights into the underlying disease biology.
3. Provide a rationale for selecting the specific five genes in the prognostic model. Explain how their functions align with COAD progression, stem cell properties, or other relevant biological processes.
4. For the 5-mRNAsi signature genes, provide more detailed information about their functions and roles in COAD. This would enhance the reader's understanding of why these genes were chosen for the prognostic model.
5. The research intent and logic of the manuscript are clear, but the English expression needs to be improved and needs to be proofread by fluent English-speaking professionals.
6.Clarify how the findings contribute to understanding COAD and potential clinical applications.

Reviewer 2 ·

Basic reporting

no comment

Experimental design

no comment

Validity of the findings

no comment

Additional comments

This study determined the prognostic genes and prognostic models associated with stem cell characterization in colon adenocarcinoma. The results of this study were validated by experiments, which were rigorous but had some shortcomings. Modify as per the comments below.
1. In the introduction, the first occurrence of an abbreviation should be indicated in full.
2. in the Raw data section, the source of the dataset should be indicated with a URL.
4. there should be more details about the analytical methodology, such a simple description will bother the reader.
a. "To calculate tumor stemness, the stemness index model was trained from the Pro genitor Cell Biology Consortium database based on OCLR (Malta et al. 2018; Wang et al. 2021). From a previously reported study, the mRNA expression-based stemness index (mRNAsi) refers to the stem cell index of each COAD sample (based on transcriptome) (Malta et al. 2018).".
b. What is Fgsea, the analytical method? Or is it an R package? And clusterProfiler, if it is an analytical method, please cite the correct literature and introduction.
c. For the drug sensitivity analysis, why were Thapsigargin, 5-Fluorouracil, Cisplatin, Gefitinib, Vinorelbine, and Paclitaxel chosen, and how were their IC50's obtained?
d. The statistical methods should all be clearly spelled out, as well as the plotting tools used.
f. In line 193, the PROGENy algorithm (Pathway RespOnsive GENes) should be spelled out in the methodology.
g. Are the statistical methods in Figure 3A-H, Figure 4B determined to be anova, or chi-square tests?
The source of the immunologic indicators in Figure 5C is not stated in the methodology. h. What is the source of the immunologic indicators in Figure 5C?
5. It is recommended that a statistic be added for the pathologic information in the three subtypes.
6. Whether there is a link between the three subtypes identified and the pathologic typing identified by the study could be added to the discussion in this section.
7. the results of GSEA could be presented in the discussion.
8. whether there is a link between the identified genes and the differential pathways should be added to this part of the discussion.
9. there are grammatical errors in a large number of statements in the article, e.g., See Table 1 for the sequences of primer pairs for the genes. needs to be corrected.

---

## Round 0.2 · Minor Revisions

Issues that need to be revised:

1. The sentence "Furthermore, to explore the oncogenic activity of different clusters in cell-specific signaling pathways such as PI3K, VEGF, EGFR, p53 and MAPK." is incomplete. It lacks a main verb and a subject. An example could be: "Furthermore, we aim to explore the oncogenic activity of different clusters in cell-specific signaling pathways such as PI3K, VEGF, EGFR, p53 and MAPK."

2. Line 141 and 142: It seems there was an editing error in this sentence. [ATCAACAGTAGCCTTTCTGAATT (HEYL-si) was the target sequence for CD83 siRNA and 142 METTL7B siRNA] should be revised to [The siRNA sequence targeting HEYL was ATCAACAGTAGCCTTTCTGAATT].
3. Please add the control siRNA sequence (si NC).

4. Immortalized colon epithelial cell line NCM460 was used as the control cell line of this study. Please demonstrate this in line 138, but not in line 236.
5. Line 202: [Firstly, T cell-inflamed GEP score to predict potential in cancer immunotherapy] should be revised to [Firstly, T cell-inflamed GEP score was used to predict potential in cancer immunotherapy.].

6. Line 206 [In addition, therapeutic signatures including gene signatures predicting radiotherapy, oncogenic pathways, and gene signatures related to targeted therapy that could shape non-inflamed TME were collected from a previous study] should be revised to [In addition, we collected therapeutic signatures from a previous study, which included gene signatures predicting radiotherapy, oncogenic pathways, and gene signatures related to targeted therapy. These signatures have the potential to shape the non-inflamed tumor microenvironment (TME).].

7. Line 234, the statement [Inhibiting the expressions of HEYL, FSTL3, FABP4, ADAM8, and EBF4] was wrong, because the authors only silenced the expression of HEYL.

8. [mRNAsi can be used as an indicator to distinguish the heterogeneity] should be revised to [mRNAsi may be used as an indicator to distinguish the heterogeneity].

9. They should focus on the role of HEYL in regulating COAD stem cell-like properties, in addition to the common malignant properties of COAD cells. This should be added in the Discussion section as a future direction.

Reviewer 1 ·

Basic reporting

no comment

Experimental design

no comment

Validity of the findings

no comment

Additional comments

The revised manuscript addresses most of the reviewers 'concerns, and I have no new comments.

Reviewer 2 ·

Basic reporting

Compared to the previous version, the manuscript has been greatly improved and the background is sufficient.

Experimental design

no comment

Validity of the findings

The author has made many supplements and fully answered my questions, but I have no new comments.

Additional comments

no comment

---

## Round 0.3 · accepted · Accept

My concerns have been addressed, and I think this revised article could be considered for publication in this journal, except that there seems to be a spelling typo at line 248 of the PDF version that needs to be corrected at the Proof stage: [C1, C2 and C2] should be revised to [C1, C2 and C3].